# FLOW MARCHING FOR A GENERATIVE PDE FOUNDATION MODEL

## ABSTRACT

Pretraining on large-scale collections of PDE-governed spatiotemporal trajectories has recently shown promise for building generalizable models of dynamical systems. Yet most existing PDE foundation models rely on deterministic Transformer architectures, which lack generative flexibility for many science and engineering applications. We propose Flow Marching, an algorithm that bridges neural operator learning with flow matching motivated by an analysis of error accumulation in physical dynamical systems, and we build a generative PDE foundation model on top of it. By jointly sampling the noise level and the physical time step between adjacent states, the model learns a unified velocity field that transports a noisy current state toward its clean successor, reducing long-term rollout drift while enabling uncertainty-aware ensemble generations . Alongside this core algorithm, we introduce a Physics-Pretrained Variational Autoencoder (P2VAE) to embed physical states into a compact latent space, and an efficient Flow Marching Transformer (FMT) that combines a diffusion-forcing scheme with latent temporal pyramids, achieving up to 15× greater computational efficiency than full-length video diffusion models and thereby enabling large-scale pretraining at substantially reduced cost. We curate a corpus of $\sim$2.5M trajectories across 12 distinct PDE families and train suites of P2VAEs and FMTs at multiple scales. On downstream evaluation, we benchmark on unseen Kolmogorov turbulence with few-shot adaptation, demonstrate long-term rollout stability over deterministic counterparts, and present uncertainty-stratified ensemble results, highlighting the importance of generative PDE foundation models for real-world applications.

## 1 INTRODUCTION

Generative models natively support sampling-based uncertainty quantification, which is critical in applications that rely on ensembles rather than single forecasts, including weather forecast (Price et al., 2024; Li et al., 2024a; Hatanpää et al., 2025), data assimilation (Bao et al., 2024b;a), machinery and materials design (Wada et al., 2023; Zeni et al., 2024), and safety margins estimation (Chen et al., 2022; Jiang et al., 2025). PDE foundation models (McCabe et al., 2024; Cao et al., 2025; Ye et al., 2025) are large, pre-trained neural operators that learn generalizable representations of spatiotemporal dynamics, enabling zero-shot prediction, control, and design across diverse physical systems. However, most foundation models for PDEs have emphasized deterministic mappings from current to future states, leaving two gaps: i) limited ability to control initial-condition (IC) uncertainty from partial/noisy observations, and ii) susceptibility to long-term rollout error accumulation.

The two gaps jointly motivate a unifying question: Can a single, scalable framework combine the efficiency and deterministic accuracy of neural operators with the generative fidelity and uncertainty modeling of diffusion methods, so the two reinforce each other to stabilize the long-term prediction through principled controlling of IC uncertainty?

We answer this with a probability-flow-based algorithm, Flow Marching (FM), which bridges a deterministic neural operator update and a stochastic flow matching step through a bridge parameter $k$. For $k = 1$, FM behaves like a neural operator, predicting the residual between two states; for $k = 0$, it reduces to a flow-matching–style sampler that generates the next state from noise; for intermediate $k$, it learns a unified velocity field that transports current state perturbed at a $k$-controlled noisy level

toward the clean next state. Hence, for any wrongly-predicted states the model should be able to guide it back to the correct dynamics, and eventually improve the long-term rollout stability.

Besides Flow Marching, a practical large-scale generative PDE foundation model is still challenging on two fronts, efficient neural architecture and data. Our efficient architecture has two components: a Pretrained-Physics Variational Autoencoder (P2VAE) and a Flow Marching Transformer (FMT). We first use P2VAE to embed original states to latent grids, avoiding prohibitive memory and computation cost of pixel-space denoising. FMT uses modern efficient modules of LLMs and further introduces a diffusion forcing scheme via a condition vector that preserves dynamical consistency without incurring full-length video self-attention, and we add a latent temporal pyramid to downsample in time for further savings. On the data side, we aggregate heterogeneous public PDE datasets (FNO-v, PDEArena, PDEBench, The Well), yielding the most comprehensive corpus used for PDE foundation models to date.

The key contributions of this paper are summarized as follows:

- The first generative PDE foundation model that unifies deterministic and stochastic modeling, which allows fast adaptation to unseen dynamics, stable long-term rollout and uncertainty-stratified ensemble generation.
- An efficient architectural design for generative modeling to dynamical systems that enables large-scale pretraining and efficacious dynamics modeling at substantially lower cost.
- A heterogeneous PDE corpus spanning FNO-v, PDEArena, PDEBench and The Well, totaling up to 233 Gigabyte and consisting of 2.5 million trajectories curated for foundation model pretraining.

## 2 RELATED WORKS

**Neural operator and PDE foundation models**    Neural operators are data-driven models that build surrogate models for science and engineering applications. Existing methods include but not limited to FNO (Li et al., 2021), DeepONet (Lu et al., 2021), PINO (Li et al., 2024b), OFormer (Li et al., 2023), and UPT (Alkin et al., 2024). They excel at fast rollout and generalization to unseen inputs (Kovachki et al., 2023; Azizzadenesheli et al., 2024; Kramer et al., 2024). As a class of specially designed neural operator, PDE foundation models based on Transformer architecture (Vaswani et al., 2017), such as ICON (Yang et al., 2023; Cao et al., 2025), MPP (McCabe et al., 2024), DPOT (Hao et al., 2024), PROSE (Liu et al., 2024; Sun et al., 2025), and PITT (Lorsung et al., 2024), learn the PDE-governed spatiotemporal dynamics through large-scale pretraining across diverse spatiotemporal systems, and enable fast adaptation to new dynamics and in-context learning ability. However, they are mostly deterministic and lack the flexibility of generative modeling.

**Diffusion-based neural PDE solver**    Diffusion-based/flow-based generative models have been explored to solve PDE equations recently (Cachay et al., 2023; Huang et al., 2024; Bastek et al., 2025; Oommen et al., 2025; Li et al., 2025), and they have been shown to outperform deterministic baselines in both predictive accuracy and physical consistency. However, these approaches adopt a paradigm of generating each new state from pure noise conditioned on the previous states, which differs fundamentally from our flow-marching strategy (see Section 3.5 for a detailed comparison) . Moreover, they are typically designed for a single dynamical system, whereas our work aims at the broader goal of building PDE foundation models.

**Flow matching**    Flow matching (FM) trains a time-dependent vector field by regressing to conditional velocities that deterministically transport white noise to data along a prescribed probability path (Lipman et al., 2023; Liu et al., 2022; Tong et al., 2024). Large-scale studies (Esser et al., 2024) further show FM can match or surpass diffusion on high-resolution image synthesis while retaining fast ODE sampling, motivating FM as a practical training principle for modern generative backbones.

**Pyramidal flow matching**    Recent work proposes Pyramidal Flow Matching (PFM) (Jin et al., 2025), which reinterprets the denoising/transport trajectory as a multi-stage spatial pyramid, where only the final stage runs at full resolution while earlier stages operate coarser and are linked through a renoising technique to preserve continuity. This yields notable training and inference efficiency

gains without losing quality (especially for video) and pairs naturally with temporal pyramids for autoregressive history compression. These ideas directly inspire our latent temporal pyramid and coarse-to-fine training/inference strategy.

**Diffusion forcing**  Diffusion Forcing (Chen et al., 2024) blends autoregressive (AR) prediction with diffusion-style denoising: a causal next-token (or next-segment) model is trained to produce future content while simultaneously denoising a set of tokens with independent per-token noise levels. Compared to pure AR, diffusion forcing lets the model get access to partially noised data distributions; compared to pure diffusion, it preserves causal structure and AR efficiency. In practice, the general concept of adopting AR for sequential data in diffusion models improves long-horizon stability (Xie et al., 2025), temporal coherence (Zhou et al., 2024), ensemble calibration (Pang et al., 2023), and condition propagation (Chen et al., 2024; Gao et al., 2024; 2025), which is suitable for general PDE dynamics modeling across heterogeneous systems.

## 3  METHODS

### 3.1  PROBLEM SETTINGS

We focus on a one-step conditional prediction process for PDE-governed spatiotemporal dynamics

$$p(\mathbf{x}_{s+1}|\mathbf{x}_{0:s}), \ s = 0, 1, \ldots \tag{1}$$

where $\mathbf{x}_s \in \mathcal{X} \subset \mathbb{R}^{H \times W \times C}$ denotes the physical state at step $s$.

A neural operator learns a time-stepping map

$$\mathbf{x}_{s+1} = \mathbf{f}_\theta(\mathbf{x}_{0:s}) \tag{2}$$

The training method is usually optimizing L2 error, which is maximum likelihood estimation (MLE) under a fixed Gaussian noise model:

$$\theta^* = \arg\min_\theta \mathbb{E}\left[\frac{1}{2\sigma^2}\|\mathbf{f}_\theta(\mathbf{x}_{0:s}) - \mathbf{x}_{s+1}\|^2\right] \Leftrightarrow \arg\max_\theta p_\theta(\mathbf{x}_{s+1}|\mathbf{x}_{0:s}) \tag{3}$$

where $p_\theta(\mathbf{x}_{s+1}|\mathbf{x}_{0:s}) = \mathcal{N}(\mathbf{f}_\theta(\mathbf{x}_{0:s}), \sigma^2\mathbf{I})$ is degenerate up to a fixed variance. Inference produces a single "deterministic" trajectory. In rollouts, errors accumulate through exponentially-grown one-step bias (see Appendix. A.1).

Instead of committing to a point estimate, we envision to model the full conditional distribution $p_\theta(\mathbf{x}_{s+1}|\mathbf{x}_{0:s})$ via a probability flow in which a learned transport field maps a noisy current state toward its successor while conditioning on the observed history. Training should regress a well-posed velocity/transport target; inference can either sample a probability-flow ODE (deterministic path) or a reverse-time SDE (stochastic sampler) to produce uncertainty-stratified ensembles. This would provide a mechanism to mitigate long-horizon drift by transporting appropriately perturbed states. We instantiate this generative conditional model with our algorithm introduced next.

### 3.2  FLOW MARCHING

Let $(\mathbf{x}_0, \mathbf{x}_1) \sim \pi$ be consecutive states of a dynamical system. We synthesize intermediate states $\mathbf{x}_t^k$ via a location-scale interpolation kernel in Fig. 1

$$\mathbf{x}_t^k = \mu_t + \sigma_t\mathbf{z}, \mu_t = t\mathbf{x}_1 + k(1-t)\mathbf{x}_0, \sigma_t = (1-t)(1-k), \mathbf{z} \sim \mathcal{N}(0, \mathbf{I}), \tag{4}$$

with $t, k \sim \text{Unif}(0, 1)$.

Conditionally,

$$q_t^k(\mathbf{x}_t^k|\mathbf{x}_0, \mathbf{x}_1, k) \sim \mathcal{N}(t\mathbf{x}_1 + k(1-t)\mathbf{x}_0, (1-t)^2(1-k)^2\mathbf{I}). \tag{5}$$

Two limits are informative: (i) $k = 0$ recovers a flow-matching kernel $\mathbf{x}_t^0 \sim \mathcal{N}(t\mathbf{x}_1, (1-t)^2\mathbf{I})$; (ii) $k = 1$ gives the deterministic neural operator interpolation $\mathbf{x}_t^1 = t\mathbf{x}_1 + (1-t)\mathbf{x}_0$. Thus, $k$ continuously bridges stochastic flow transport and deterministic operator learning.

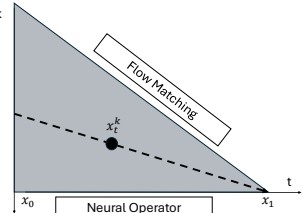

Figure 1: Location-scale interpolation kernel for flow marching

Using the equivalent form

$$\mathbf{x}_t^k = \mathbf{x}_0 + t(\mathbf{x}_1 - \mathbf{x}_0) - (1-t)(1-k)(\mathbf{x}_0 - \mathbf{z}). \tag{6}$$

Differentiation gives the sample-wise velocity

$$\mathbf{u}_t^k = \frac{d}{dt}\mathbf{x}_t^k = (1-k)(\mathbf{x}_0 - \mathbf{z}) + \mathbf{x}_1 - \mathbf{x}_0 = \frac{\mathbf{x}_1 - \mathbf{x}_t^k}{1-t}. \tag{7}$$

For an isotropic Gaussian $q_t^k(\mathbf{x}) = \mathcal{N}(\mu_t, \sigma_t^2 \boldsymbol{I})$, its conditional score at the sampled point $\mathbf{x} = \mathbf{x}_t^k$ is

$$\nabla_{\mathbf{x}} \log q_t^k(\mathbf{x}_t^k) = -\frac{\mathbf{x}_t^k - \mu_t}{\sigma_t^2} = -\frac{\mathbf{z}}{(1-t)(1-k)}. \tag{8}$$

We substitute Eq. 8 into Eq. 7, and get the score-velocity decomposition:

$$\mathbf{u}_t^k = (\mathbf{x}_1 - k\mathbf{x}_0) + (1-t)(1-k)^2 \nabla_{\mathbf{x}} \log q_t^k(\mathbf{x}_t^k). \tag{9}$$

Therefore, the transport velocity is a well-posed learnable target: its random part is aligned with an intrinsic geometric direction of the conditional density (the score), and its deterministic part is fixed by the pair $(\mathbf{x}_0, \mathbf{x}_1)$ and the bridge parameter $k$. Regressing a model $\mathbf{g}$ to $\mathbf{u}_t^k$ theoretically recovers the posterior-mean transporting field $\mathbf{g}^*$ (see Appendix B).

From a score matching perspective, to approximate $\mathbf{u}_t^k$ by $\mathbf{g}_\theta$ would suggest feeding $(\mathbf{x}_t^k, k, t)$ as input because $q_t^k$ is a parametrized by $(k, t)$. However, we notice that including $k$ empirically slows convergence and can induce a failure mode that an intermediate denoised state $\tilde{\mathbf{x}}_t$ may drift off the nominal bridge $(k, t)$, so the provided $k$ no longer matches the state, yielding biased gradient signals . We therefore adopted the equivalent frame-interpolation target $\mathbf{u}_t^k = (\mathbf{x}_1 - \mathbf{x}_t^k)/(1-t)$ in Eq. 7, so that $(\mathbf{x}_t^k, t)$ is sufficient. This $k$-free objective can also be intuitively understood as the linear vector pointing to the end state $\mathbf{x}_1$ from the current state $\mathbf{x}_t^k$. This frame-interpolation view makes the training interface minimal: once $\mathbf{x}_t^k$ is constructed offline, the supervision depends only on $(\mathbf{x}_1 - \mathbf{x}_t^k)$. In the inference process, the training objective can effectively reduce long-term rollout error accumulation, because wrongly-predicted states $\mathbf{x}_1$ can be interpreted as $\mathbf{x}_1^{\hat{k}}$ with $(1 - \hat{k})$ IC uncertainty in the next prediction step, which can still be guided towards correct $\mathbf{x}_2$ without requiring $\hat{k}$ as an input.

The form is numerical stiff near $t \to 1$. We therefore precondition the target by $(1-t)$ and obtain the flow marching objective:

$$\mathcal{L}_{\text{FM}} = \frac{1}{2}\mathbb{E}\left[||(1-t)\mathbf{g}_\theta(\mathbf{x}_t^k, t) - (\mathbf{x}_1 - \mathbf{x}_t^k)||^2\right]. \tag{10}$$

A comprehensive numerical analysis on the error accumulation comparison between deterministic neural operator and our flow marching scheme is provided in Appendix. A.2.

### 3.3 CONDITIONAL FLOW MARCHING THROUGH DIFFUSION FORCING

There exists a variety of different dynamics in the training dataset. To accommodate these dynamics in one PDE foundation model, we introduce the conditional form of flow marching and design an approach to condition the dynamics through past states.

The conditional flow marching target could be derived from a conditional probability:

$$q_t^k(\mathbf{x}, \mathbf{h}) = \mathbb{E}_{(\mathbf{x}_0, \mathbf{x}_1)}\left[q_t^k(\mathbf{x}|\mathbf{x}_0, \mathbf{x}_1, \mathbf{h})\right], \quad \mathbf{h} = \mathbf{h}_s. \tag{11}$$

where $\mathbf{h}_s$ summarizes the observed history $\mathbf{x}_{0:s}$. Following diffusion forcing (DF) (Chen et al., 2024), we maintain a filtered latent state that evolves as

$$\mathbf{h}_s \sim p_\phi(\mathbf{h}_s|\mathbf{h}_{s-1}, \mathbf{x}_{s,t_s}^{k_s}, t_s), \tag{12}$$

implemented by a lightweight RNN with parameters $\phi$. By induction, $\mathbf{h}_s = \mathbf{h}_s(\mathbf{x}_{0:s})$ acts as a learned sufficient statistics for the (approximately) time-stationary PDE dynamics.

Given $\mathbf{h}_s$, the conditional flow marching target and objective become

$$\mathcal{L}_{\text{CFM}} = \frac{1}{2} \mathop{\mathbb{E}}_{\substack{t_s, \mathbf{x}_s, \mathbf{x}_{s+1}, \mathbf{h}_s \\ \mathbf{h}_s \sim p_\phi(\mathbf{h}_s|\mathbf{h}_{s-1}, \mathbf{x}_{s,t_s}^{k_s}, t_s)}} \sum_{s=1}^{T} \left[\left\|(1 - t_s)\mathbf{g}_\theta(\mathbf{x}_{s,t_s}^{k_s}, t_s, \mathbf{h}_{s-1}) - (\mathbf{x}_{s+1} - \mathbf{x}_{s,t_s}^{k_s})\right\|^2\right], \tag{13}$$

where $\mathbf{x}_{s,t_s}^{k_s} = \mathbf{x}_s + t_s(\mathbf{x}_{s+1} - \mathbf{x}_s) - (1 - t_s)(1 - k_s)(\mathbf{x}_s - \mathbf{z})$, $t_s, k_s$ are independently sampled at each physical timestep $s$.

Crucially, integrating out the auxiliary variables $(\mathbf{z}, k, t)$ and conditioning on $\mathbf{h}_s$, recovers a learned one-step predictive law:

$$p_\theta(\mathbf{x}_{s+1}|\mathbf{x}_{0:s}) \approx \iiint p_\theta(\mathbf{x}_{s+1}|\mathbf{x}_{s,t}^k, t, \mathbf{h}_s)p(\mathbf{z})p(k)p(t)d\mathbf{z}dkdt, \quad \mathbf{h}_s = F_\phi(\mathbf{x}_{0:s,t_{0:s}}^{k_{0:s}}, t_{0:s}). \tag{14}$$

Under universal function approximation and optimal training of $(\theta, \phi)$, integral along the flow marching path driven by $\mathbf{g}_\theta(\cdot, \cdot, \mathbf{h}_s)$ yields a Monte-Carlo sample from this conditional distribution. Thus, the DF latent $\mathbf{h}_s$ and $\mathcal{L}_{\text{CFM}}$ realize a consistent estimator of the transition kernel $p_\theta(\mathbf{x}_{s+1}|\mathbf{x}_{0:s})$, closing the loop between history filtering and uncertainty-aware transport fields.

### 3.4 LATENT TEMPORAL PYRAMIDS

We introduce two techniques to improve the computational efficiency of our model. Firstly, we introduce P2VAE parametrized by $\omega$ to embed the state $\mathbf{x}$ to latent space $\mathbf{y}$, borrowing the idea from LDM (Rombach et al., 2022) to balance computation cost and detail preservation. The training of P2VAE is separated from and before the training of FMT.

$$\mathbf{y} = \mathcal{E}_\omega(\mathbf{x}), \quad \hat{\mathbf{x}} = \mathcal{D}_\omega(\mathbf{y}),$$
$$\mathcal{L}_{\text{VAE}} = \frac{1}{2}\mathbb{E}\|\mathbf{x} - \hat{\mathbf{x}}\|^2 + \beta D_{\text{KL}}(q_\omega(\mathbf{y}|\mathbf{x})||p(\mathbf{y})). \tag{15}$$

To further simplify the computational complexity, we introduce temporal pyramids in PFM (Jin et al., 2025), which resonates with the fact that a physics dynamical system is mostly Markovian and prediction relies less on farther previous states. For early $s$, we use downsampled latent states to propagate the PDE condition $\mathbf{h}_s$. In practice, we always train the conditional flow marching model on 4 consecutive states $(\mathbf{x}_0, \mathbf{x}_1, \mathbf{x}_2, \mathbf{x}_3)$, where FMT calculates the flow marching objective $\mathcal{L}_{\text{CFM}}$ with latent temporal pyramids $(Down(\mathbf{y}_{0,t_0}^{k_0}, 8), Down(\mathbf{y}_{1,t_1}^{k_1}, 4), Down(\mathbf{y}_{2,t_2}^{k_2}, 2), \mathbf{y}_{3,t_3}^{k_3})$ as input.

### 3.5 PLUG-AND-PLAY UNCERTAINTY QUANTIFICATION PROCESSES

While $(k_0, k_1, k_2, k_3) = (1, 1, 1, 1)$ is a deterministic prediction setting of our framework, which is applied in scenarios where accurate predictions are required, such as long-term rollout, our model also provides test-time knobs to model both IC uncertainty (via the bridge parameter $k$) and aleatoric uncertainty (via SDE-based sampling) without retraining.

Conventional generative conditional forecasters (Price et al., 2024; Oommen et al., 2025) implement $\mathbf{x}_{s+1} = G_\theta(\boldsymbol{\epsilon}, \mathbf{h}_s)$ with $\mathbf{h}_s = F_\phi(\mathbf{x}_{0:s})$ fixed; sampling seeds vary only $\text{Var}(\mathbf{x}_{s+1}|\mathbf{x}_{0:s}, \theta)$ which is the aleatoric uncertainty. They lack a mechanism to dial the trust in $\mathbf{x}_s$ and thus cannot isolate or control IC uncertainty without replacing the input history by posterior samples. Even the diffusion forcing (Chen et al., 2024) fails to provide a principled IC uncertainty-driven ensemble because the prediction is modeled by $p_\theta(\mathbf{x}_{s+1}|\mathbf{h}_s(\hat{\mathbf{x}}_s, \mathbf{x}_{0:s-1}))$, and modifying $\mathbf{x}_s$ would entangle the aleatoric

uncertainty and IC uncertainty within the $\mathbf{h}_s$. In contrast, our flow marching process provides an explicit knob to express IC uncertainty via the bridge parameter $k_3$ without influencing $\mathbf{h}_{s-1}(\mathbf{x}_{0:s-1})$. To simulate IC uncertainty, we use the Euler ODE sampler (discretization on $t$) to propagate an intermediate state $\mathbf{x}_t^k$ to $\mathbf{x}_1$ through the probability flow velocity $\mathbf{g}_\theta$. $(t_0, t_1, t_2, t_3)$ are initialized to be 0, and are updated simultaneously during the flow marching process. The discretization is taken to be $N = 100$ throughout the evaluation phase, with $dt = 0.01$. We set $(k_0, k_1, k_2)$ to be 1 and $k_3$ less than 1. Given that the smaller the $k$, the larger the uncertainty about the current state, this setting allows $\mathbf{h}_3$ to be passed down from a clean history, and generate possible $\mathbf{x}_4$ out of gradually added IC uncertainty. $(k_0, k_1, k_2)$'s parametrization choice can be further explored.

Aleatoric variability remains available by rewriting the flow-marching kernel as an SDE sampler. To simulate aleatoric uncertainty, we recover reverse-time SDE out of PF-ODE following the formula provided in Flow-GRPO (Liu et al., 2025). Details of derivation can be found in Appendix. C.

$$dx = \left[ \mathbf{g}_\theta + \frac{1}{2}\eta^2(1-t)(\mathbf{x} - \mathbf{x}_s - t\mathbf{g}_\theta) \right] dt + \sigma(t)d\bar{\mathbf{w}}_t, \; \sigma_t = \eta(1-t) \tag{16}$$

For epistemic uncertainty, we can utilize methods such as MC-Dropout (Gal & Ghahramani, 2016), yet a full epistemic evaluation is out of scope for this paper.

## 4 EXPERIMENTS

### 4.1 SETUP

**Dataset gathering** We consider a combination of public benchmark datasets for PDE foundation models: FNO-v (Li et al., 2021), PDEBench (Takamoto et al., 2024), PDEArena (Gupta & Brandstetter, 2022), and the Well (Ohana et al., 2025) to form a heterogeneous dataset consisting of 12 distinct dynamical systems. All the dynamical systems are 2D intrinsically, and three physical fields are chosen at maximum. We compressed the aforementioned datasets to the format of $128 \times 128$ spatial resolution with 3 multiphysics channels (c3p128) with float16 precision to form a 233 GB dataset, consisting over 2.5M trajectories with length 4. We provide the exact compression ratio and dataset information in Appendix D.

**Training dataset** The heterogeneous dataset is partitioned into train, valid, and test sets according to the original settings of each sub-dataset first; under the cases where original partition doesn't exist, we use a ratio of 8:1:1. We train P2VAE and FMT on the train set. Datasets are sampled with equal probabilities according to the practice in DPOT (Hao et al., 2024). P2VAE's AdamW optimizer is used with $\beta_1 = 0.9$ and $\beta_2 = 0.995$, cosine learning rate schedule with 10% of linear warm up, and a weight decay of 1e-4; FMT's AdamW optimizer is used with $\beta_1 = 0.9$ and $\beta_2 = 0.95$, cosine learning rate schedule with 10% of linear warm up, and a weight decay of 0.01. Base learning rates of 1e-4 for a 256 batch size are adjusted linearly to batch sizes and inverse square proportional to model sizes to balance convergence speed and training stability. We conduct a two stage training recipe. We trained 2 P2VAEs, 16M and 87M, for 100k steps with KL term's weight $\beta =$ 1e-3. Based on the 16M P2VAE (with frozen weights), we train 3 FMTs with size 6M, 42M, and 138M (Small, Base, and Large) on the same training dataset for another 100k steps.

**Evaluation metrics** To assess the reconstruction and prediction quality of our model, we employ both the L2 relative error (L2RE), which is a common practice of PDE foundation models, and the variance-normalized root mean square error (VRMSE), as suggested by Ohana et al. (2025).

**Implementation details** For P2VAE, we reuse the standard SD-VAE (Rombach et al., 2022) architecture to compress each state from c3p128 to c16p16 ($12\times$ compression rate) following the recommendation by Hansen-Estruch et al. (2025). P2VAE-16M uses 64 as the base dimensions, while P2VAE-87M uses 128. For FMT, we use the AdaLN-Zero mechanism introduced in (Peebles & Xie, 2023) to condition a SiT (Ma et al., 2024). In the Transformer side, we adopt the modern architecture RMSNorm and SwiGLU introduced by Llama-2 (Touvron et al., 2023). Multi-head self-attention with head dim 64 is implemented with FlashAttention v2 (Dao, 2023). FMT-S, FMT-B, and FMT-L have 256, 512, and 768 as the embedding dimensions, respectively. The RNN in the diffusion forcing scheme is a GRU (Chung et al., 2014) which shares the same internal dimension

as the embedding dimension in SiT; the current states are compressed onto a single token by cross attention to update the latent state $\mathbf{h}$ to inform dynamics.

## 4.2 EFFICIENCY

Compared to a vanilla video-diffusion model (Ho et al., 2022) adaptation to our setting that operates with bidirectional self-attention across 4 frames with 256 tokens each and quadratic attention complexity, the efficiency gain due to FMT could be estimated by

$$\eta = \frac{(4 \times 16^2)^2}{(2^2)^2 + (4^2)^2 + (8^2)^2 + (16^2)^2} = 15. \tag{17}$$

## 4.3 BASELINES

Baseline methods include: UNet (Ronneberger et al., 2015), FNO (Li et al., 2021), CNextU-net (Ibtehaz & Kihara, 2023), which are trained on individual dynamics; DPOT (Hao et al., 2024), MPP (McCabe et al., 2024), VICON (Cao et al., 2025), which are PDE foundation models jointly trained on several dynamics. All of the above is based on a deterministic neural operator setting. The results are listed in Tab. 1. The entries with $*$ is the implementation provided in the Well benchmark (Ohana et al., 2025). We provide the reconstruction error in L2RE and VRMSE of our P2VAEs to demonstrate the compression loss level due to the autoencoder structure for further comparisons. Note that since we unified the sub-datasets to p128c3 and float16, the metrics taken from other papers could be different on our format-unified dataset; specifically, the PA-NS, PA-NSC, PB-CNSL, PB-CNSH are only compressed through precision change without resolution change, which allows fair comparisons with statistics mentioned in previous works .

Table 1: P2VAE reconstruction error compared to benchmark PDE models. The best (or better) results among the existing statistics are in bold.

| L2RE | FNO-v5 | FNO-v4 | FNO-v3 | PA-NS | PA-NSC | PA-SWE | PB-CNSL | PB-CNSH | PB-SWE | W-AM | W-GS | W-SWE | W-RB | W-SF | W-TR | W-VE |
|---|---|---|---|---|---|---|---|---|---|---|---|---|---|---|---|---|
| UNet | 0.198 | 0.119 | 0.0245 | 0.102 | 0.337 | | 0.463 | 0.313 | 0.0521 | | | | | | | |
| FNO | 0.116 | 0.0922 | 0.0156 | 0.210 | 0.384 | | 0.153 | 0.130 | 0.00912 | | | | | | | |
| DPOT-30M | **0.0553** | **0.0442** | **0.0131** | 0.0991 | 0.316 | | **0.0153** | **0.0245** | **0.00657** | | | | | | | |
| MPP-116M | | | | 0.0617 | | | 0.164 | 0.209 | | | | | | | | |
| VICON-88M | | | | 0.111 | | | 0.1561 | 0.0597 | | | | | | | | |
| P2VAE-16M | 0.0890 | 0.0850 | 0.124 | 0.0651 | 0.0604 | 0.1093 | 0.0267 | 0.0334 | 0.438 | 0.0466 | 0.0774 | 0.0629 | 0.105 | 0.0956 | 0.0401 | 0.0360 |
| P2VAE-87M | 0.0802 | 0.0732 | 0.115 | **0.0582** | **0.0527** | **0.1039** | 0.0266 | 0.0325 | 0.186 | **0.0329** | **0.0400** | **0.0596** | **0.0802** | **0.0846** | **0.0374** | **0.0274** |
| VRMSE | FNO-v5 | FNO-v4 | FNO-v3 | PA-NS | PA-NSC | PA-SWE | PB-CNSL | PB-CNSH | PB-SWE | W-AM | W-GS | W-SWE | W-RB | W-SF | W-TR | W-VE |
| UNet* | | | | | | | | | | 0.2489 | 0.2252 | 0.3620 | 1.4860 | 3.447 | 0.2418 | 0.4185 |
| FNO* | | | | | | | | | | 0.3691 | **0.1365** | 0.1727 | 0.8395 | 1.189 | 0.5001 | 0.7212 |
| CNextU-net* | | | | | | | | | | **0.1034** | 0.1761 | 0.3724 | 0.6699 | 0.8080 | **0.1956** | 0.2499 |
| P2VAE-16M | 0.2240 | 0.2457 | 0.2721 | 0.0936 | 0.0850 | 0.1135 | 0.6028 | 0.3386 | 0.6504 | 0.4064 | 0.4916 | 0.1126 | 0.2499 | 0.1718 | 0.2838 | 0.2962 |
| P2VAE-87M | **0.1886** | **0.2192** | **0.1986** | **0.0828** | **0.0743** | **0.1074** | **0.4444** | **0.2714** | **0.2945** | 0.2016 | 0.3298 | **0.0951** | **0.1886** | **0.1453** | 0.2324 | **0.1568** |

## 4.4 VISUALIZATION

Sampled trajectories generated with FMT-L-138M on test sets from all 12 distinct PDE systems are displayed in Appendix. E.

## 4.5 DOWNSTREAM EVALUATION RESULTS

**Adapting foundation model to isotropic Kolmogorov turbulence**  According to REPA-E (Leng et al., 2025), we finetune the pretrained model (P2VAE-16M and FMT-B-42M) to adapt to an unseen system with a stop-gradient operation after the generation of latent states $\mathbf{y}$, so that the conditional flow marching loss won't deteriorate the autoencoder. The end-to-end finetuning loss is derived as

$$\mathcal{L}(\theta, \phi, \omega) = \mathcal{L}_{\text{CFM}}(\theta, \phi) + \lambda_{\text{VAE}}\mathcal{L}_{\text{VAE}}(\omega). \tag{18}$$

We conduct the experiment on an isotropic Kolmogorov turbulence dataset with $u$ and $v$ fields at $Re = 222$ (Sardar & Skillen, 2025). We finetuned our FMT-B-42M model on 200 of the training trajectories in the train set for 5k steps with $\lambda_{\text{VAE}} = 1$ and test the performance on 500 trajectories in the test set. We compare the finetuning result with training from scratch (Scratch) as well. Details are provided in Appendix. F. The metrics are shown in Table. 2, and one exemplary vorticity ($\boldsymbol{\omega} = \frac{\partial v}{\partial x} - \frac{\partial u}{\partial y}$) reconstruction and prediction case is shown in Fig. 2.

Table 2: Few-shot adaptation result on the Kolmogorov turbulence dataset. P2VAE-16M-FT and FMT-B-42M-FT denote two finetuned model; P2VAE-FMT-B denotes the joint model trained from scratch.

| Model | L2RE | VRMSE |
|---|---|---|
| P2VAE-16M-FT | 0.0243 | 0.0614 |
| FMT-B-42M-FT | 0.0836 | 0.1053 |
| Scratch | 0.1342 | 0.2367 |

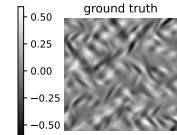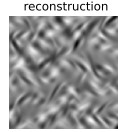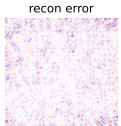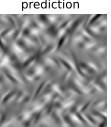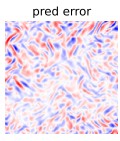

Figure 2: Reconstructed and predicted vorticity by the finetuned model

**Long-term rollout**   We test the long-term rollout performance of our model on the PDEArena-NS, PDEBench-CNS-Low, and PDEBench-CNS-High datasets, and make the comparison with the statistics in VICON (Cao et al., 2025), as shown in Tab. 3. We plot sample trajectories based on FMT-B-42M in Appendix G.

Table 3: Comparison of long term rollout errors (in L2RE), with best results in bold.

| L2RE | Case | FMT-S-6M | FMT-B-42M | FMT-L-138M | VICON-88M |
|---|---|---|---|---|---|
| Step 1 | PA-NS | 0.1060 | 0.0879 | **0.0745** | 0.1110 |
| | PB-CNS-Low | 0.0960 | 0.0796 | **0.0557** | 0.1561 |
| | PB-CNS-High | 0.0890 | 0.0450 | **0.0411** | 0.0597 |
| Step 5 | PA-NS | 0.1889 | 0.1355 | **0.1292** | 0.2300 |
| | PB-CNS-Low | 0.0957 | 0.0958 | **0.0872** | 0.2456 |
| | PB-CNS-High | 0.1091 | 0.0992 | **0.0797** | 0.1973 |
| Step 10 | PA-NS | 0.3318 | 0.2234 | **0.2088** | 0.3618 |
| | PB-CNS-Low | 0.1444 | 0.1119 | **0.1004** | 0.3747 |
| | PB-CNS-High | 0.1621 | 0.1218 | **0.1198** | 0.5788 |
| Last step | PA-NS | 0.5664 | 0.6134 | **0.5271** | 0.7781 |
| | PB-CNS-Low | 0.2820 | **0.1298** | 0.1311 | 0.3903 |
| | PB-CNS-High | 0.2130 | 0.1392 | **0.1279** | 0.7117 |
| Average | PA-NS | 0.4176 | 0.3859 | **0.3048** | 0.5627 |
| | PB-CNS-Low | 0.1480 | 0.1035 | **0.0960** | 0.2708 |
| | PB-CNS-High | 0.1497 | 0.1092 | **0.0914** | 0.3006 |

In DPOT (Hao et al., 2024), the authors noticed that injecting noise during training would benefit the long-term rollout capability because the distribution of model-predicted states can be partially taken into account. However, they lack a systematic way to evaluate the noise level, which in turn demands a hyperparameter tuning process. In contrast, our model can deal with any noisy state because the distribution $q_t^k, k \in [0, 1]$ has been modelled, so that any misaligned predicted states are exposed during the training implicitly, which in turn minimize the exposure bias (Arora et al., 2023; Huang et al., 2025) during long term prediction.

**Generate uncertainty-stratified ensemble of next states**   For both IC uncertainty-driven ensemble and aleatoric uncertainty-driven ensemble, we sampled single trajectory from PDEArena-NS and tested it on the FMT-B-42M model to generate 32-batch size ensembles.

For IC uncertainty-driven ensemble, by tuning bridge parameter $k_3$ during the generation, we can effectively generate an ensemble of possible next state given a noisy initialization $k_3\mathbf{x}_3 + (1 - k_3)\mathbf{z}$ and a concluded PDE condition $\mathbf{h}_3$ from clean past frames $(\mathbf{x}_0, \mathbf{x_1}, \mathbf{x_2})$. The variance of the predicted ensemble is a decreasing function of prior noise level $k_3$, which is displayed with selected generated samples at different $k_3$ in Fig. 3. For aleatoric uncertainty-driven ensemble, we tune the $\eta$ in Eq. 16, and present the generated samples in Appendix H. Both the IC and aleatoric uncertainty-driven ensembles show no artifacts of the flow field.

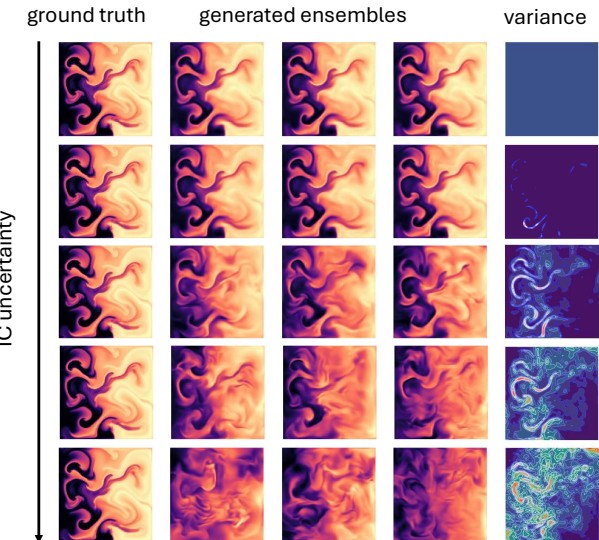

Figure 3: Generated ensembles at different $k_3$: $1, 0.8, 0.6, 0.4, 0.1$ (from top to bottom).

## 5 CONCLUSION

In this paper, we propose a conditional flow marching algorithm with a diffusion forcing scheme to construct a generative PDE foundation model that predicts future states given a series of past states. Empowered by a diverse training dataset, it displays excellent few-shot adaptation performance on unseen isotropic Kolmogorov turbulence. While deterministic neural operators learn a time-stepping map and yields a degenerate conditional distribution, our generative formulation targets the full conditional $p_\theta(\mathbf{x}_{s+1}|\mathbf{x}_{0:s})$ via flow marching objective, and predictions are produced by integrating PF-ODE or sampling by reverse-time SDE without extra training. This provides principled aleatoric uncertainty and IC uncertainty and a noise-aware stable rollout process, while retaining the efficiency of operator-style updates in latent space.

We envision the current generative model to serve as a foundational tool for PDE-related applications that have a real-world impact. In the future, on the architecture side, we expect advanced Transformer models to enable better convergence of flow marching target; also we expect a stronger autoencoder would unlock the base performance bottleneck by minimizing the compression loss. On the application side, various IC uncertainty could be explored including regional blurring, low resolution inputs, etc.; more applications could be explored based on the current model setting, including but not limited to data assimilation, sparse reconstruction, equation inference, computational design, etc.

### ETHICS STATEMENT

Our contributions enable advances in generative modeling application in PDE dynamics field. This has the potential to significantly impact a wide range of downstream applications. While we do not anticipate specific negative impacts from this work, there is potential for misuse. We encourage the research community to consider the ethical implications and potential dual-use scenarios when applying these technologies in sensitive domains.

## REPRODUCIBILITY STATEMENT

We have made significant efforts to ensure the reproducibility of our work. All model architectures, training hyperparameters, and evaluation protocols are described in detail in Sections 3, 4 of the main paper. Complete proofs of the theoretical results are included in Appendices. To facilitate reproduction of our results, we plan on making a GitHub repo publicly available which contains the training and evaluation scripts, and environment specifications; the preprocessed dataset should also be shared under the same repository.

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

## A  NUMERICAL ANALYSIS ON ERROR ACCUMULATION

### A.1  ERROR ACCUMULATION IN DETERMINISTIC NEURAL OPERATORS

**Setup**  Let the true one-step dynamics be $\Phi : \mathcal{X} \to \mathcal{X}$ and the learned neural operator be $f_\theta : \mathcal{X} \to \mathcal{X}$. Rollouts are generated by

$$\mathbf{x}^*_{n+1} = \Phi(\mathbf{x}^*_n), \; \mathbf{x}_{n+1} = f_\theta(\mathbf{x}_n) \tag{A1}$$

where $\mathbf{x}^*_s$ denotes the true states, $\mathbf{x}_s$ denotes the states on simulated trajectory. Define at test time

$$\delta_n := \mathbf{x}_n - \mathbf{x}^*_n, \; e_n := f_\theta(\mathbf{x}_n) - \Phi(\mathbf{x}_n) \tag{A2}$$

**Error analysis**

$$\delta_{n+1} = f_\theta(\mathbf{x}_n) - \Phi(\mathbf{x}^*_n) = e_n + (\Phi(\mathbf{x}_n) - \Phi(\mathbf{x}^*_n)). \tag{A3}$$

Assume that $\Phi$ is $L$-Lipschitz, then

$$\|\delta_{n+1}\| \le \|e_n\| + L\|\delta_n\|, \tag{A4}$$

with induction / Gronwall

$$\|\delta_n\| \le L^n \|\delta_0\| + \sum_{j=0}^{n-1} L^{n-1-j} \|e_j\|. \tag{A5}$$

$L$ is an intrinsic property of true dynamics $\Phi$. If $L > 1$, the error accumulates geometrically for both $\|\delta_0\|$ and $\|e_j\|$; under the cases where $L \le 1$, $\|e\|$ still contributes to the error growth.

A neural network $\theta$ is optimized through L2 objective

$$\arg\min_\theta \mathbb{E}\|f_\theta(\mathbf{x}) - \Phi(\mathbf{x})\|^2 \tag{A6}$$

such that we can assume a relative accuracy on $f_\theta$

$$\|e_n\| = \|\epsilon_{\text{train}}(\mathbf{x}_n)\| \le \rho\|\Phi(\mathbf{x}_n)\|, \forall n \tag{A7}$$

and assume $\|\delta_0\| = 0$, so that

$$\|\delta_n\| \le \rho \sum_{j=0}^{n-1} L^{n-1-j} \|\Phi(\mathbf{x}_j)\| \le \rho \sum_{j=0}^{n-1} L^{n-1-j} D_{\max}, \; D_{\max} = \sup \|\Phi(\mathbf{x})\|. \tag{A8}$$

### A.2  ERROR ACCUMULATION FOR FLOW MARCHING AND COMPARISONS WITH DETERMINISTIC NEURAL OPERATORS

**Setup**  The learned probability flow velocity gives

$$\dot{\mathbf{x}} = \mathbf{g}_\theta(\mathbf{x}, t), \; \mathbf{g}_\theta = \mathbf{g}^* + r, \tag{A9}$$

where $r$ is the residual function. The next state $\mathbf{x}^*_{s+1}$ and $\mathbf{x}_{s+1}$ is obtained through integrating PF-ODE

$$\mathbf{x}^*_s(t) := \mathbf{x}^*_s + \int_0^t \mathbf{g}^*(\mathbf{x}^*_s(t'), t')dt', \; \mathbf{x}_s(t) := \mathbf{x}_s + \int_0^t \mathbf{g}_\theta(\mathbf{x}_s(t'), t')dt', \tag{A10}$$

and

$$\mathbf{x}^*_{s+1} = \mathbf{x}^*_s(1), \; \mathbf{x}_{s+1} = \mathbf{x}_s(1). \tag{A11}$$

Define at test time (assuming integration along PF-ODE doesn't introduce error)

$$\delta_s(t) := \mathbf{x}_s(t) - \mathbf{x}^*_s(t), \; \delta_s(0) = \delta_s, \; \delta_s(1) = \delta_{s+1}. \tag{A12}$$

**Error analysis** Start by differentiate $\delta_s(t)$ along $t$

$$\dot{\delta}_s(t) = \mathbf{g}_\theta(\mathbf{x}_s(t), t) - \mathbf{g}^*(\mathbf{x}_s^*(t), t) = \mathbf{g}^*(\mathbf{x}_s(t), t) - \mathbf{g}(\mathbf{x}_s^*(t), t) + r(\mathbf{x}_s(t), t). \quad \text{(A13)}$$

Since $\mathbf{g}^*(\mathbf{x}, t) = (\mathbf{x}_1 - \mathbf{x})/(1 - t)$, we have

$$\mathbf{g}^*(\mathbf{x}_s(t), t) - \mathbf{g}(\mathbf{x}_s^*(t), t) = \frac{1}{1-t}(\mathbf{x}_s^*(t) - \mathbf{x}_s(t)) = -\frac{1}{1-t}\delta_s(t). \quad \text{(A14)}$$

We further express $r(\mathbf{x}_s(t), t)$ as a linearized form of $\delta_s(t)$

$$r(\mathbf{x}_s(t), t) = r(\mathbf{x}_s^*(t), t) + R_s(t)\delta_s(t), \ R_s(t) := \int_0^1 \nabla_\mathbf{x} r(\mathbf{x}_s^*(t) + \xi\delta_s(t), t)d\xi \quad \text{(A15)}$$

and arrive at an inhomogeneous linear ODE $\forall s$

$$\dot{\delta}(t) = (R(t) - \frac{1}{1-t})\delta(t) + r(\mathbf{x}^*, t). \quad \text{(A16)}$$

Change its varaible to $y(t) := \delta(t)/(1 - t)$ to avoid singularity near $t \to 1$, we have

$$\dot{y}(t) = R(t)y(t) + \frac{r(\mathbf{x}^*, t)}{1-t}. \quad \text{(A17)}$$

Let $\Psi(t, \sigma)$ be the state-transition operator of $R(\cdot)$, it has the solution

$$y(t) = \Psi(t, 0)y(0) + \int_0^t \Psi(t, \sigma)\frac{r(\mathbf{x}^*(\sigma), \sigma)}{1-\sigma}d\sigma \quad \text{(A18)}$$

Assume that $r(\cdot, t)(1-t)$ is $L$-Lipschitz wrt $\mathbf{x}$ (because $\mathbf{g}_\theta$ is trained against the $\|(1-t)\mathbf{g}_\theta - (\mathbf{x}_1 - \mathbf{x})\|^2$ target, such that

$$\|R(t)\| \le L(t)/(1-t), \|\Psi(\sigma, \sigma)\| \le \exp(\int_\sigma^t \frac{L(\xi)}{1-\xi}d\xi) \quad \text{(A19)}$$

which gives

$$\|\delta(t)\| \le (1-t)\exp(\int_0^t \frac{L(\xi)}{1-\xi}d\xi)\|\delta(0)\| + (1-t)\int_0^t \frac{\exp(\int_\sigma^t \frac{L(\xi)}{1-\xi}d\xi)}{1-\sigma}\|r(\mathbf{x}^*, \sigma)\|d\sigma. \quad \text{(A20)}$$

Also

$$\|r(\mathbf{x}^*, t)\| = \frac{\|\epsilon_{\text{train}}(\mathbf{x}^*, t)\|}{1-t} \quad \text{(A21)}$$

Hence, we have the control over $\|\delta(t)\|$ that

$$\|\delta(t)\| \le a_t\|\delta(0)\| + b_t, a_t = (1-t)\exp(\int_0^t \frac{L(\xi)}{1-\xi}d\xi), b_t = (1-t)\int_0^t \frac{\exp(\int_\sigma^t \frac{L(\xi)}{1-\xi}d\xi)}{(1-\sigma)^2}\|\epsilon_{\text{train}}(\mathbf{x}^*, \sigma)\|d\sigma \quad \text{(A22)}$$

Up to the last discretization step during the integrator, we left a residual time $\epsilon$, so that

$$a = \epsilon\exp(\int_0^{1-\epsilon} \frac{L(\xi)}{1-\xi}d\xi), b = \epsilon\int_0^{1-\epsilon} \frac{\exp(\int_\sigma^{1-\epsilon} \frac{L(\xi)}{1-\xi}d\xi)}{(1-\sigma)^2}\|\epsilon_{\text{train}}(\mathbf{x}^*, \sigma)\|d\sigma \quad \text{(A23)}$$

Similar to neural operator case, we have a relative accuracy on $\|\epsilon_{\text{train}}\|$ such that

$$\|\epsilon_{\text{train}}\| \le \rho\|\mathbf{x}_1 - \mathbf{x}\| \quad \text{(A24)}$$

so

$$a \le \epsilon^{1-L}, b \le -\rho\epsilon^{1-L}\log\epsilon\|\mathbf{x}_1 - \mathbf{x}\| \quad \text{(A25)}$$

for $L < 1$ (can be achieved through optimization), optimally pick $\epsilon = \exp(-1/(1-L))$, so

$$a = e^{-1} < 1, b = \frac{\rho}{1-L}e^{-1}\|\mathbf{x}_1 - \mathbf{x}\| \quad \text{(A26)}$$

such that the induction form yields

$$\|\delta_n\| \le \frac{\rho}{1-L}e^{-1}\sum_{j=0}^{n-1}e^{-n+1+j}\|\mathbf{x}_{j+1} - \mathbf{x}\| \le \frac{\rho}{(1-L)(e-1)}D_{\max}, D_{\max} = \sup\|\Phi(\mathbf{x}) - \mathbf{x}\|. \quad \text{(A27)}$$

**Error comparison**  To conclude, we compare Eq. A27 with Eq. A8, and find out that while deterministic neural operator form is governed by a summation of un-optimizable $\Phi$ intrinsic Lipschitz constant, the probability-based flow marching form can be reached with optimizable Lipschitz constant to be below 1, and arriving at a constant upper bound.

## B  DERIVATIONS FOR POSTERIOR MEAN $\mathbf{g}^*$ AND CONTINUITY EQUATION FOR LOCATION-SCALE INTERPOLATION KERNEL

We mentioned that regressing $\mathbf{g}$ to $\mathbf{u}_t^k = (\mathbf{x}_1 - \mathbf{x}_t^k)/(1 - t)$ would recover the posterior-mean transporting field $\mathbf{g}^*$. This can be given by the standard L2-regression lemma

$$\arg\min_\theta \mathbb{E}\left[\|\mathbf{g}_\theta(\mathbf{x}_t^k, t) - u_t^k\|^2\right] = \mathbb{E}\left[\mathbf{u}_t^k|\mathbf{x}_t^k, t\right] \tag{B1}$$

proved through

$$\mathbb{E}_{(\mathbf{x},t)}\left[\mathbb{E}\left[\|\mathbf{g}(\mathbf{x}, t) - \mathbf{u}\|^2\right]|\mathbf{x}, t\right] = \mathbb{E}_{(\mathbf{x},t)}\left[\|\mathbf{g}(\mathbf{x}, t) - \mathbb{E}\left[\mathbf{u}|\mathbf{x}, t\right]\|^2 + \mathrm{Var}(\mathbf{u}|\mathbf{x}, t)\right] \tag{B2}$$

where $\mathrm{Var}(u|x, t)$ doesn't depend on $\mathbf{g}$, so the minimizer is $\mathbf{g}(\mathbf{x}, t) = \mathrm{RHS}$ almost everywhere.

Then we can derive the continuity equation based on the previous equation, so that $\mathbf{g}^*$ indeed transports the mixture $q_t$.

Define the mixture marginal

$$q_t(\mathbf{x}) = \mathbb{E}_{(\mathbf{x_0},\mathbf{x_1},k)}\left[q_t^k(\mathbf{x}|\mathbf{x_0}, \mathbf{x_1})\right]. \tag{B3}$$

For any smooth test function $\phi$,

$$\frac{d}{dt}\mathbb{E}_{q_t}\left[\phi(\mathbf{x}_t^k)\right] = \mathbb{E}_{q_t}\left[\nabla\phi(\mathbf{x}_t^k) \cdot \mathbf{u}_t^k\right] = \int \nabla\phi(\mathbf{x}) \cdot \underbrace{\mathbb{E}\left[\mathbf{u}_t^k \mid \mathbf{x}_t^k = \mathbf{x}, t\right]}_{:= \mathbf{g}^*(\mathbf{x},t)} q_t(\mathbf{x})d\mathbf{x} \tag{B4}$$

Integrating by parts gives

$$\frac{d}{dt}\int \phi(\mathbf{x})q_t(\mathbf{x})d\mathbf{x} = -\int \phi(\mathbf{x})\nabla_\mathbf{x} \cdot (q_t(\mathbf{x})\mathbf{g}^*(\mathbf{x}, t))\, d\mathbf{x}, \tag{B5}$$

because $\int_{\partial\Omega} \phi(\mathbf{x})\mathbf{g}^*(\mathbf{x}, t)q_t(\mathbf{x})n(\mathbf{x})d\mathbf{S} = 0, \phi \in \mathcal{C}_c^\infty(\Omega)$. Since this holds for all $\phi$, $q_t$ and $\mathbf{g}^*$ satisfy the continuity equation

$$\partial_t q_t(\mathbf{x}) + \nabla_\mathbf{x} \cdot (q_t(\mathbf{x})\mathbf{g}^*(\mathbf{x}, t)) = 0, \tag{B6}$$

The location-scale interpolation kernel is admissible – it induces a path of densities $q_t$ that is transported by the velocity field equal to the posterior mean $\mathbf{g}^*$ of the sample-wise velocities $\mathbf{u}_t^k$; smoothness and integrability are valid.

## C  REVERSE-TIME SDE ADAPTED FROM PF-ODE

**Probability-flow ODE**  The trained transport field $\mathbf{g}_\theta(\mathbf{x}, t, \mathbf{h}_s)$ satisfies the continuity equation

$$\partial_t q_t(\mathbf{x}|\mathbf{x}_{0:s}) + \nabla_\mathbf{x} \cdot (q_t(\mathbf{x}|\mathbf{x}_{0:s})\mathbf{g}_\theta(\mathbf{x}, t, \mathbf{h}_s)) = 0, \tag{C1}$$

so the deterministic sampler is simply

$$\frac{d\mathbf{x}}{dt} = \mathbf{g}_\theta(\mathbf{x}, t, \mathbf{h}_s). \tag{C2}$$

**Transform PF-ODE to reverse-time SDE**  For any diffusion schedule $\sigma(t) \geq 0$, an equivalent family of reverse-time SDEs shares the same marginals $q_t$ as the PF-ODE when the score is exact:

$$d\mathbf{x} = \left[\mathbf{g}_\theta - \frac{1}{2}\sigma(t)^2\nabla_\mathbf{x}\log q_t(\mathbf{x}|\mathbf{x}_{0:s})\right]dt + \sigma(t)d\bar{\mathbf{w}}_t. \tag{C3}$$

From the frame-interpolation target $(1 - t)\mathbf{g}_\theta \approx \mathbf{x}_{s+1} - \mathbf{x}$, define the predicted endpoint and bridge mean

$$\tilde{\mathbf{x}}_{s+1} := \mathbf{x} + (1 - t)\mathbf{g}_\theta(\mathbf{x}, t, \mathbf{h}_s), \quad \hat{\boldsymbol{\mu}}_t := \mathbf{x}_s + t(\tilde{\mathbf{x}}_{s+1} - \mathbf{x}_s). \tag{C4}$$

A consistent plug-in score approximate is (take $k = 1$)

$$\hat{\mathbf{s}}(\mathbf{x}, t, \mathbf{h}_s) := -\frac{\mathbf{x} - \hat{\boldsymbol{\mu}}_t}{\hat{\sigma}(t)^2} = -\frac{(1-t)(\mathbf{x} - \mathbf{x}_s - t\mathbf{g}_\theta)}{\hat{\sigma}(t)^2} \tag{C5}$$

where $\hat{\sigma}$ is an approximation to ground truth variance

$$\hat{\sigma}(t) = \sqrt{(1-t)^2(1-k)^2 + \epsilon(1-t)^2} \tag{C6}$$

where $\epsilon$ is a finite a-priori lower bound on unresolved variance in the score approximation, preventing the drift-correction term blow up as well. Insert $\hat{\mathbf{s}}$ into the SDE term, so that the reverse-time SDE adapted from PF-ODE is

$$d\mathbf{x} = \left[\mathbf{g}_\theta + \frac{1}{2}\frac{\sigma(t)^2}{\hat{\sigma}(t)^2}(1-t)(\mathbf{x} - \mathbf{x}_s - t\mathbf{g}_\theta)\right]dt + \sigma(t)d\bar{\mathbf{w}}_t, \sigma(t) = \eta(1-t). \tag{C7}$$

Take $k = 1$ when measuring aleatoric uncertainty, still $(\sigma/\hat{\sigma})^2$ is finite; $\sigma$ has the form

$$\sigma(t) = \eta(1-t) \tag{C8}$$

and

$$d\mathbf{x} = \left[\mathbf{g}_\theta + \frac{1}{2}\frac{\eta^2}{\epsilon^2}(1-t)(\mathbf{x} - \mathbf{x}_s - t\mathbf{g}_\theta)\right]dt + \sigma(t)d\bar{\mathbf{w}}_t, \sigma(t) = \eta(1-t). \tag{C9}$$

**Discretization and Euler-Maruyama integrator**

$$d\mathbf{x} = \left[\mathbf{g}_\theta + \frac{1}{2}\frac{\eta^2}{\epsilon^2}(1-t)(\mathbf{x} - \mathbf{x}_s - t\mathbf{g}_\theta)\right]dt + \eta(1-t)\sqrt{dt}\boldsymbol{\epsilon}_t, \ \boldsymbol{\epsilon}_t \sim \mathcal{N}(0, \boldsymbol{I}) \tag{C10}$$

where $d\mathbf{x}$ and $dt$ will be correspondingly discretized to $\Delta\mathbf{x}$ and $\Delta t$, with $\boldsymbol{\epsilon}_t$ being sampled each $\Delta t$, to form a Euler-Maruyama integrator. Without lost of generality, we take $\epsilon$ to be 1.

# D   DATASET DESCRIPTION

All the data are compressed to float16 (half) precision to enable the Data Distributed Parallel training on a 4 H-100 GPU node.

**FNO-v**   We upsampled original data from c1p64 to c3p128 (the 2nd and 3rd dimension are filled with zero). The dataset size is expanded from 11.1GB to 21GB. Trajectory count: FNO-v5 – 15.4k, FNO-v4 – 368k, FNO-v3 – 184k.

**PDEArena**   For the PDEArena-NavierStokes(PA-NS) and PDEArena-NavierStokesCond(PA-NSC), the dataset size is compressed from 60GB to 25GB. For the PDEArena-ShallowWaterEquation(PA-SWE), it was slightly expanded to 62GB from 76.6GB because additional all-zero channels are provided. Trajectory count: PA-NS – 48k, PA-NSC – 120k, PA-SWE – 470k.

**PDEBench**   For the PDEBench-CompressibleNavierStokes(PB-CNS), unimportant physical fields are filtered.  Thus, it becomes 65GB compressed from 551GB. For the PDEBench-ShallowWaterEquation(PB-SWE), it is compressed to 0.3GB from 6.2GB, the 2nd and 3rd dimension is filled with zero. Trajectory count: PB-CNS – 598k, PB-SWE – 77.6k.

**The Well**   For the Well-GrayScott(W-GS), we fill the 3rd dimension with zero, ending up with a 5.3GB data set compressed from 153GB. For the Well-ActiveMatter(W-AM), we downsampled the data from c3p256 to c3p128, and obtained a compressed 1.1GB dataset from 51.3GB. For the Well-PlanetShallowWaterEquation(W-SWE), we downsampled the data from c3p256,512 to c3p128 and filtered out unimportant fields, so the data size is compressed to 9.3GB from 185.8GB. For the Well-RayleighBenard(W-RB), we downsampled the data from c3p512,128 to c3p128, and get a 26GB dataset from 342GB original data. For the Well-ShearFlow(W-SF), it is compressed to 14GB from 547GB by filtering out unimportant fields. For the Well-TurbulentRadiativeLayer2D(W-TR), it is downsampled from c3p128,384 to c3p128, thus compressed to 0.5GB from 6.9GB. For the Well-ViscoElasticInstability(W-VE), it is downsampled from c3p512 to c3p128, thus compressed to 0.5GB from 66GB. Trajectory count: W-GS – 92.2k, W-AM – 13.4k, W-SWE – 96.4k, W-RB – 266.6k, W-SF – 175.6k, W-TR – 7k, W-VE – 5.3k.

# E    VISUALIZATION FOR TEST SETS OF 12 PDE SYSTEMS

FNO-v: sampled trajectories are displayed in Fig. E1.

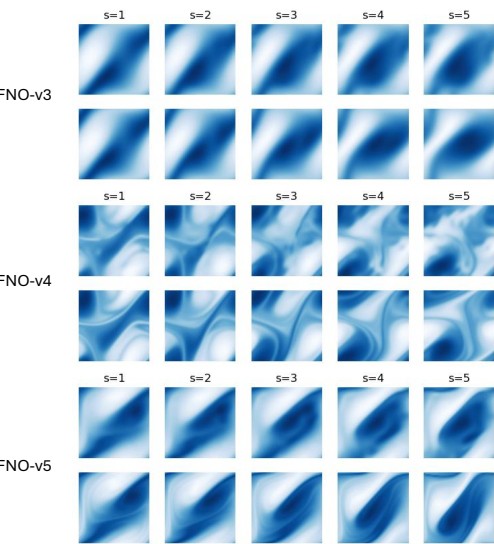

Figure E1: Sampled trajectories from FNO-v3, FNO-v4, FNO-v5. Upper row: prediction. Bottom row: ground truth.

PDEArena: sampled trajectories are displayed in Fig. E2.

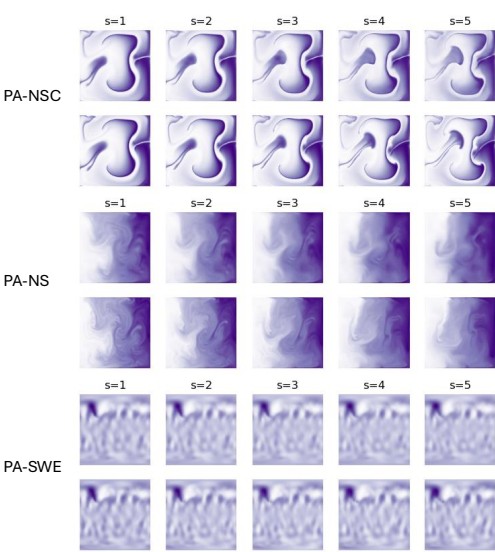

Figure E2: Sampled trajectories from PA-NSC, PA-NS, PA-SWE. Upper row: prediction. Bottom row: ground truth.

PDEBench: sampled trajectories are displayed in Fig. E3.

The Well: sampled trajectories are displayed in Fig. E4.

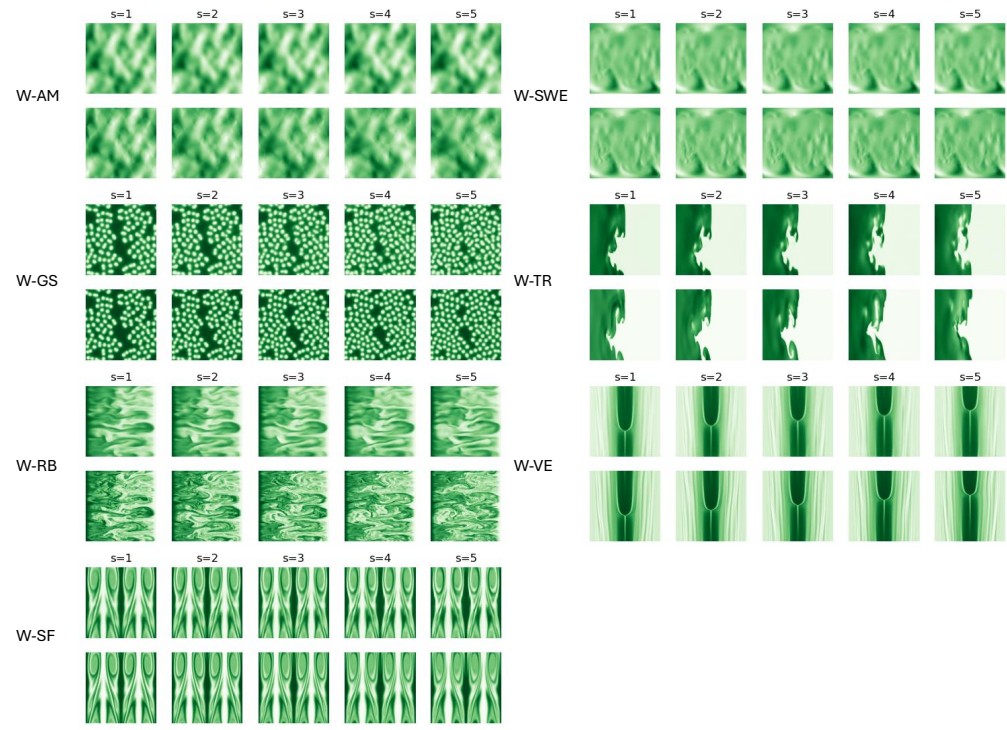

Figure E3: Sampled trajectories from PB-CNSH, PB-CNSL, PB-SWE. Upper row: prediction. Bottom row: ground truth.

Figure E4: Sampled trajectories from W-AM, W-GS, W-RB, W-SF, W-SWE, W-TR, W-VE. Upper row: prediction. Bottom row: ground truth.

## F  TRAINING FROM SCRATCH ON UNSEEN SYSTEMS

The Scratch model is trained following the same protocol as the finetuning process, including the stop-gradient operation and hyper parameters. We train the model to 50k steps to analyze the results, and the loss curve is provided in Fig. F1.

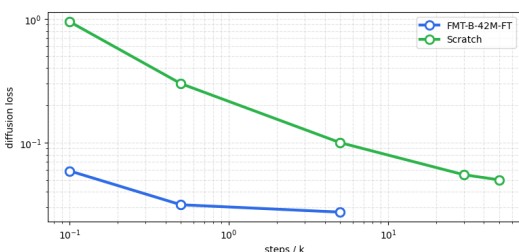

Figure F1: Convergence plot of finetuning and training from scratch on the unseen Kolmogorov turbulence system.

## G    ROLLOUT VISUALIZATIONS

Sampled long-term rollout trajectories are provided in Fig. G1 (P-NS), Fig. G2 (PB-CNS-L) and Fig. G3 (PB-CNS-H).

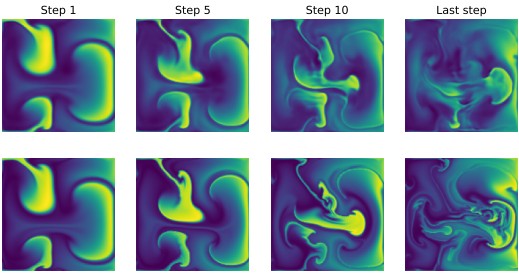

Figure G1: Sampled long-term rollout trajectories from PDEArena-NS by FMT-B-42M. Upper row: prediction. Bottom row: ground truth.

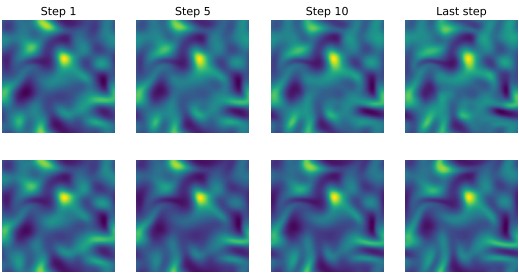

Figure G2: Sampled long-term rollout trajectories from PDEBench-CNS-Low by FMT-B-42M. Upper row: prediction. Bottom row: ground truth.

## H    GENERATED UNCERTAINTY-STRATIFIED ENSEMBLES

We provide the scalar variance of IC uncertainty-driven ensembles in Fig. H1.

We also provide the aleatoric uncertainty-driven ensembles and their variance in Fig. H2.

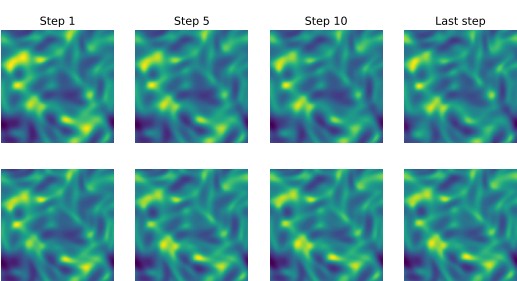

Figure G3: Sampled long-term rollout trajectories from PDEBench-CNS-High by FMT-B-42M. Upper row: prediction. Bottom row: ground truth.

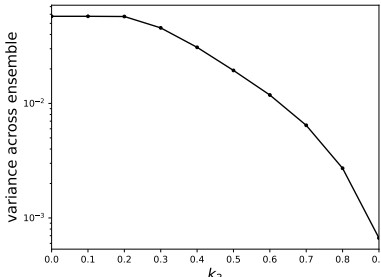

Figure H1: Average of batch-wise variation of $\mathbf{x}_4$ IC uncertainty-driven ensemble generated at different $\mathbf{x}_3$ noise levels $k_3$.

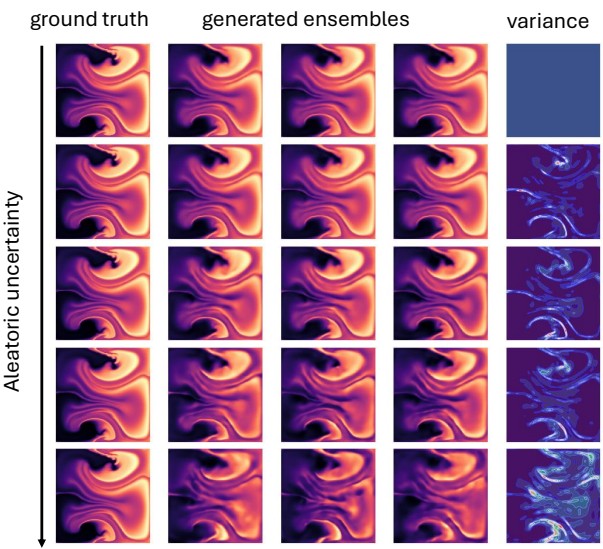

Figure H2: Generated ensembles at different $\eta$: $0, 0.1, 0.4, 0.7, 1.0$ (from top to bottom).

