# OpenReview forum: "Flow Marching for a Generative PDE Foundation Model"
_ICLR.cc/2026/Conference — ICLR 2026 Conference Withdrawn Submission_

### Official Review · Reviewer_AJHs · 2025-10-27

**Soundness:** 2
**Presentation:** 2
**Contribution:** 2
**Rating:** 4
**Confidence:** 2

**Summary:**

This paper presents a novel framework for building generative foundation models for partial differential equations (PDEs). The authors propose ​​Flow Marching (FM)​​, an algorithm that bridges neural operator learning with flow matching to enable uncertainty-aware ensemble generation while reducing long-term rollout drift. The method is supported by a ​​Physics-Pretrained Variational Autoencoder (P2VAE)​​ for latent space compression and an efficient ​​Flow Marching Transformer (FMT)​​ architecture. The model is pretrained on a large-scale corpus of ~2.5M trajectories across 12 PDE families and demonstrates strong performance in few-shot adaptation, long-term rollout stability, and uncertainty quantification.

**Strengths:**

- Generative Flexibility​​: The FM algorithm natively supports uncertainty-aware ensembles, addressing a key limitation of deterministic PDE foundation models.
- ​​Large-Scale Validation​​: Experiments span 12 PDE systems, demonstrating few-shot adaptation to unseen dynamics and long-term rollout stability.

**Weaknesses:**

- Incomplete Benchmarking:​​ The experimental evaluation, particularly in Table 1, presents an incomplete picture. While the results show that the proposed method is outperformed by the DPOT-30M model on several datasets (e.g., FNO-v5/v4/v3 and PB-CNSL/CNSH/SWE),  the performance of DPOT on the Well dataset is missing.  A fair and comprehensive benchmark requires that all major baselines are evaluated across the same set of test datasets. Similarly, the long-term rollout analysis in Table 3 compares the proposed method only against VICON. To better situate the claimed improvements in long-term stability, comparisons with other relevant foundation models  would significantly strengthen the validity of the results.
- Unclear Visual Explanation:​​ Figure 1, which illustrates the core "location-scale interpolation kernel," is not sufficiently explained in the main text. The caption is terse and fails to guide the reader on how to interpret the diagram to understand the bridging process between flow matching and neural operator. A more detailed description in the caption, coupled with a clearer explanation in Section 3.2, is necessary to make this foundational concept accessible.

**Questions:**

See weaknesses.

---

### Official Review · Reviewer_9Hu6 · 2025-10-28

**Soundness:** 1
**Presentation:** 2
**Contribution:** 2
**Rating:** 2
**Confidence:** 4

**Summary:**

The authors present a method for training a model that can interpolate between deterministic, next-step prediction and a flow-matching process. The authors pretrain the model on a large corpus of data and benchmark against other models.

**Strengths:**

- Developing a framework to interpolate between deterministic and probabilistic prediction is interesting and is well-motivated by IC uncertainty.
- The developed framework is elegant and, on paper, seems to fulfill the requirements posed by the authors (aleatoric/IC uncertainty, diffusion forcing, etc.)
- The authors make an effort to pretrain on a large corpus of data

**Weaknesses:**

I think the method is interesting and could be promising, but the major concern is that the evaluations are not rigorous and careless to the point that many of the results cannot be fairly interpreted:
- In Table 1, the experimental setup for many of the baselines is not even the same as the tested setup, and furthermore, the setup of each baseline is not even consistent with other baselines or made clear. I had to look at the original papers (DPOT, MPP, The Well) to find the error numbers that were copied over and the setup that was used to get these errors.
    - Setup 1: DPOT is run with a context length of 10 (takes 10 prior frames to make 1 future prediction) and the error is averaged over 5 autoregressive predictions. The Unet in the upper portion of the table is also setup like this. Extra channels are padded and inputs are sampled to a common, 128x128 resolution.
     - Setup 2: Unet/FNO/CNextUnet in the lower portion of the table is setup with a context length of 1 and makes 1 future prediction, and the error is only at the next timestep. The resolution of the input/outputs are variable, and can be rectangular (512x128 RBC). Furthermore, these inputs are not downsampled like P2VAE.
         - The FFNO baseline from the Well is missing (not a huge problem, but why only copy over errors from three out of the four models?)
     - Setup 3: The P2VAE uses a common, 128x128 input, with extra channels discarded. The evaluation is a reconstruction error on a single frame, not a prediction error, which is a different task than the other baselines considered.
     - There are more baselines cited that I did not attempt to try to figure out how these error numbers were generated. Even then, the reported setups are surface-level, there are many more parameters (num epochs, training compute, model size) that are not easily accessible yet should be made standardized or at least known to readers so that they themselves can make a fair comparison.

- The issue stems from the fact that the authors decide to re-use baseline metrics from prior works, without making an attempt to adapt their own evaluation to match. Furthermore, when taking numbers from multiple prior works, each prior work uses a different training scheme, model size, data downsampling, error calculation, etc. which causes numerous inconsistencies when comparing results across different papers, not to mention comparing your model to these numbers.
- My recommendation is to focus on a few, well-motivated experiments and baselines models and run the baselines yourself to ensure that the setups are consistent across all runs. Without this, it is very challenging to actually figure out if your method is doing better, and more broadly, may mislead casual readers about the true nature of what is going on.

As a result of the lack of rigor in evaluations, many of the results are not convincing:
- The main table (Table 1) compares a reconstruction task to a prediction task, which is misleading. Reconstruction is much easier than prediction (both from prior work: https://arxiv.org/abs/2507.02608, and shown in Table 2 (VAE vs FMT losses)) and even then, the autoencoder does not seem to do better than some predictive baselines (although I am unsure if we can even compare this from what is noted in previous section). If the autoencoder is already worse, then it is likely that the FMT model will be worse than a majority of the baselines considered, especially when the FMT is trained with the 16M variant of the autoencoder.
- In Table 3, we get an idea of the model’s rollout/prediction performance, but it is only compared to a single baseline. It is known that Unet is a good model for regular gridded benchmark problems, so a comparison to FNO/UNet would be good to make.

There are also some minor concerns:
- The strategy used to downsample data is a little curious:
    - Discarding extra physical variables and labeling them as “unimportant” is rather dismissive. For example, the point of solving compressible Navier Stokes is to get access to new physical variables that are not present in incompressible Navier Stokes, such as energy.
    - Similarly, downsampling physical fields destroys finer-scale features that could be important. Especially in turbulent systems like Rayleigh-Benard Convection, there is mixing behavior that is likely lost when downsampling.
- The runtime of FMT is longer than any of the baselines (100 denoising steps), compared to most baselines being a direct prediction (1-step).

Overall, I think this is a promising idea, but needs substantial work to fix most of the experiments. It seems that you have put a lot of effort into the work, so I am sorry about the harsh review, but these are basic concerns about experimental rigor. Please let me know if I have had any misunderstandings about the nature of the experiments.

**Questions:**

- What is k set to for experiments in Table 3?
- How does the model performance compare to pure flow matching?
   - In order to motivate the need for flow marching, it should be shown that it is better than the deterministic case and purely flow matching case.

---

### Official Review · Reviewer_E8HE · 2025-10-29

**Soundness:** 3
**Presentation:** 2
**Contribution:** 3
**Rating:** 2
**Confidence:** 4

**Summary:**

The work proposes a PDE foundation model based on flow matching deformation. The method seems novel and has theoretical basis, but the explanation of the method is unclear and the experiment lacks analysis. Please refer to Weaknesses for details. The writing of the paper needs to be clearer, which is an important basis for improving rating.

**Strengths:**

1. The proposed method is based on flow matching, with theoretical guarantees, and has been proven to reduce error accumulation. Although the relationship between the proof and error accumulation is currently unclear, further detailed explanation is needed by the author.
2. A comprehensive dataset was constructed and experimentally validated on multiple PDE systems, although it is currently unclear how the experiments and comparisons were conducted as the experimental section is too rough.

**Weaknesses:**

1. In Line 21, there is a typo.
2. In Line 48. This solution requires more explanation of the reasons why generating models can reduce the accumulation of errors in rollouts? For flow matching, it is also conditional generation, and the initial conditions have a significant impact on flow matching, similar to deterministic models.
3. In related works, it lacks many of the latest generative model-based works, such as FunDiff.
4. There are many independent points in the methods section, including the derivation of flow marching, P2VAE, etc., lacking an understanding of the overview, including their connections and relationships.
5. Why is it named flow marching? It's too similar to flow matching and there's no clear explanation.
6. The idea in sec. 3.2 looks interesting, but please explain the motivation behind it intuitively in the text.
7. In sec. 3.2, is t the time step of the PDE system or the step in flow matching? It looks similar to the number of steps in flow matching, but what is the relationship between this number of steps and the accumulation of errors over time?
8. In Line 200, why does it reduce error accumulation need to be explained in detail here?
9. In sec. 3.3, it is necessary to distinguish between physical time and flow matching steps. It looks like physical time here because there is history.
10. How to ensure that the hidden state of FM output and VAE training do not undergo distribution transition?
11. Line 247, a typo.
12. The proposed method is a variation of flow matching, why is there no flow matching baseline? This should be a strong baseline to verify the introduced k parameter.
13. I don't understand the meaning of Table 1 and need a detailed caption. Why are there missing parts?
14. The experimental section is too rough and lacks discussion and analysis. For example, why is DPOT in Table 1 better on some data?
15. Why use a table in Table 2? Use curves for clearer demonstration.

**Questions:**

See the weaknesses.

---

### Official Review · Reviewer_RQeV · 2025-10-30

**Soundness:** 2
**Presentation:** 1
**Contribution:** 2
**Rating:** 2
**Confidence:** 4

**Summary:**

This paper proposes Flow Marching, an algorithm that bridges neural operator learning and flow matching. In addition, the authors introduce a Physics-Pretrained Variational Autoencoder to embed states into a latent space. Also, they combine a diffusion-forcing scheme with latent temporal pyramids. The experiments are across 12 distinct PDE families with downstream evaluation.

**Strengths:**

- The proposed method bridges deterministic neural operator learning and stochastic flow matching.
- The paper introduces several techniques to improve efficiency.
- This model is trained on a large-scale dataset.

**Weaknesses:**

- Some related works about the foundation diffusion/flow-based model are not included, such as [Fundiff: Diffusion models over function spaces for physics-informed generative modeling], [FluidZero: Mastering Diverse Tasks in Fluid Systems through a Single Generative Model].
- l219: The meaning of '$h_s$ summarizes the observed history' is not clear. What does 'summarize' mean? Also, definitions of $t_s, k_s$ should be after Eq.12.
- Since the algorithm contains many parts and is not clear, it would be more readable if there were an architecture illustration of the method.
- The baselines are suggested to include the diffusion/flow-based neural operator trained on single tasks, such as [Wavelet Diffusion Neural Operator], [Dyffusion: A dynamics-informed diffusion model for spatiotemporal forecasting].
- In Table 1, the improvements compared with baselines are limited. Compared with other methods, only 7/16 results are the best (bold). And DPOT only contains 30M parameters, but outperforms P2VAE-87M a lot.
- There are no comparisons with other methods in Figure 2, which makes this figure meaningless.
- There are no baselines in Section 4.5, which makes performance not comparable with existing methods.
- Typo: The period is missing in the caption of Figure 1 and Figure 2.

**Questions:**

- There is $h_s$ in the probability of expectation (buttom of $\mathbb{E}$), but only $h_{s-1}$ in the term that takes the expectation (right of $\mathbb{E}$). Is there a typo?
- Why is there an approximately equal sign in Eq. 14? Does this mean that this equation only holds approximately?
- The input of $F_ϕ$ in Eq. 14 contains $x^{k_{0:s}}_{0:s,t_{0:s}}$, which should be computed with $x_{s+1}$, but $x_{s+1}$ is the target of the prediction. So how can we get $x_{s+1}$ during sampling?

---

### Note · Authors · 2025-11-14

I have read and agree with the venue's withdrawal policy on behalf of myself and my co-authors.